# Adolescent abortion care trajectories and safety in Ethiopia, Malawi, and Zambia: A comparative mixed methods study

Ernestina Coast[1]*, Tamara Fetters[2], Malvern Tatenda Chiweshe[3], Bellington Vwalika[4], Risa Griffin[5], Luke Tembo[6], Joe Strong[1], Charlotte Chishiba[6], Malede Birara[7], Grace Chiudzu[8], Abrham Getachew[9], Samuel Muluye Welelaw[10], Godfrey Kangaude[11], Nyovani Madise[12]

1 Department of International Development, London School of Economics and Political Science, London, United Kingdom, 2 Ipas, Chapel Hill, North Carolina, United States of America, 3 Critical Studies in Sexualities and Reproduction, Rhodes University, Makhanda, South Africa, 4 University of Zambia, Department of Obstetrics and Gynaecology, Lusaka, Zambia, 5 Risa Griffin Consulting LLC, Washington, District of Columbia, United States of America, 6 Ipas Malawi, Lilongwe, Malawi, 7 Department of Obstetrics and Gynaecology, St Paul's Hospital Millennium Medical College, Addis Ababa, Ethiopia, 8 Department of Obstetrics and Gynaecology, University of Malawi, Lilongwe, Malawi, 9 St Paul's Hospital Millennium Medical College, School of Public Health, Addis Ababa, Ethiopia, 10 Ipas Ethiopia, Addis Ababa, Ethiopia, 11 Critical Studies in Sexualities and Reproduction, Rhodes University, Makhanda, South Africa, 12 Global Health Consultant & African Institute for Development Policy, Lilongwe, Malawi

* e.coast@lse.ac.uk

## Abstract

Lack of access to safe abortions continues to be a major sexual and reproductive health concern. Adolescents can face barriers to safe abortions due to the unique implications of their age. Understanding adolescent abortion experiences and care trajectories is critical. Qualitative and quantitative evidence is analysed from interviews with 313 adolescents aged 10–19 years who sought abortion-related care in public health facilities in Addis Ababa (Ethiopia), Lilongwe (Malawi), and Lusaka (Zambia) between April 2018 and September 2019. The trajectories framework is used to understand how adolescent abortion-related care-seeking differs across a range of socio-legal national contexts. A comparative study design incorporates varying levels of restriction on access to abortion: Ethiopia (legal on broad social or economic grounds, services implemented); Zambia (legal on broad social or economic grounds, complex services with barriers to implementation and information provision); and Malawi (legally highly restricted). Most adolescents (n = 97, 98%) in Ethiopia obtained a medically safe abortion, and most adolescents (n = 70, 64%) in Zambia and almost all adolescents (n = 94, 94%) in Malawi obtained a less medically safe abortion. There is a significant association between country and whether an adolescent tried to end a pregnancy before coming to the facility, $X^2(2, N = 313) = 135.93$, p < 0.001. In Malawi 97% (n = 101) of adolescents had sought to end the pregnancy before coming to the facility, compared to 18% (n = 18) in Ethiopia. Cross-country

**Data availability statement:** The data can be accessed via: https://reshare.ukdataservice.ac.uk/856965/.

**Funding:** This work was supported by the Medical Research Council (MR/P011454/1 to EC). The funders had no role in study design, data collection and analysis, decision to publish, or preparation of the manuscript.

**Competing interests:** The authors have declared that no competing interests exist.

variations in the relative safety of the abortion and type of care sought by adolescents reflect national laws, policies and service availability. The most facility-based abortions were recorded where care is most accessible (Ethiopia), and most non-facility-based and least safe abortions were recorded where care is the most restricted (Malawi). Across all countries, adolescents experienced delays to their care-seeking; 39% (n = 39), 71% (n = 74), and 66% (n = 73) in Ethiopia, Malawi, and Zambia respectively. Adolescents reported high levels of respectful treatment across countries, with a minority reporting negative experiences. A minority of adolescents in each country reported that unofficial money was paid to facility staff. There were different experiences of post-abortion contraception in the three countries, including an absence of choice. Adolescents' trajectories – particularly those involving multiple attempts and barriers to achieving abortion care – require tenacity and determination. Our analyses show that less restrictive abortion policies and accessible abortion services improve adolescent chances to access abortion care.

## Introduction

Access to safe abortion is influenced by overlapping structural factors - legal, political, socio-cultural, and economic – that operate concurrently and in ways that are often hidden from view. While abortion is becoming safer over time, abortion is least safe in the most legally restrictive contexts [1]. Globally, in the least restrictive contexts, 87% of abortions are classified as safe; in the most restrictive contexts, only a quarter of abortions are classified as safe [2]. Three quarters (75.6%) of abortions in Africa as a region are classified as unsafe, the majority of which are least safe [1]. Adolescents are more likely to experience the negative consequences of unsafe abortion, with around 70% of hospitalizations due to unsafe abortion in Africa being among adolescents below 20 years of age [3]. In addition to structural factors, an individual's ability to access a safe abortion in any context is affected by various intersecting factors, including those that are gendered, classed, racialised, aged, and dis/abled, compounding the challenges they face in making autonomous reproductive decisions. Understanding in detail the different ways in which someone seeks abortion care offers insights into these individual- and structural-level factors of abortion access.

Despite some progress in adolescent sexual and reproductive health and rights (ASRHR), adolescents' (10–19 years) access to safe abortion remains limited and inequitable. While global and regional commitments recognise the specific SRH needs of adolescents and the lifecourse implications of (not) meeting them [4–6], they often stop short of referring to proven health interventions, including access to safe abortion. A review of essential healthcare services offered in low- and middle-income countries showed that only some adolescent SRH services were included, mainly related to contraception and sexually transmitted infections or the human immunodeficiency virus (HIV) [7].

Adolescence involves multiple transitions (physical, cognitive, social, and economic), and evolving capabilities, experiences, and behaviours during adolescence can have lifelong consequences [8–11]. Meeting the needs of sexually active adolescents or adolescent survivors of sexual violence who want to have an abortion is critically important for health-, rights- and justice-related outcomes [12–13]. Evidence about younger (below 15 years) adolescents' abortion care-seeking behaviours is limited [14–15]. The exclusion of very young adolescents from most abortion research is stigmatising; it ignores their experiences and circumstances [16]. Where evidence on abortion rates for sexually active adolescents in Africa are available, they are higher relative to older women [17–19]. Adolescents are less likely to access abortion care than older people for reasons such as: lower levels of knowledge or constrained agency to act on their knowledge [20], later pregnancy recognition [21], fewer financial resources, higher likelihood of delaying care-seeking [22–25], lower ability to navigate health systems [26], and higher levels of stigma [27].

Research highlights the need for evidence to understand how people obtain abortions and the safety of these abortions, especially in legally restrictive settings [28]. Accessing abortion care is time-critical; it can be complex, non-linear, and protracted, or it can be simple, straightforward, and short. The conceptual framework of abortion trajectories [29] situates abortion care-seeking in a comprehensive and multi-level framework. A care-seeking trajectory is not assumed to be linear, and occurs within a structural (health, legal, socio-cultural, political) environment. It focuses on the individual at the centre of care-seeking and uses the concept of the abortion trajectory starting from pregnancy awareness to abortion-related care. In addition to informing the World Health Organization's *Abortion Care Guideline* [30–31], the framework has been used in diverse abortion research [32–34]. We used this framework to compare adolescent abortion care-seeking and safety in three contrasting socio-legal contexts in Africa: Ethiopia, Malawi, and Zambia. We comparatively analyse trajectories of adolescent abortion care-seeking, including delays and experiences of care.

## Context

The three countries were purposively selected for varying levels of restriction on access to abortion, particularly for adolescents: in Ethiopia, abortion is legal on broad economic or social grounds and services are implemented; in Zambia, abortion is legal on broad economic or social grounds, but there are multiple barriers to information and implementation making access complex; in Malawi, legal abortion is highly restricted (Table 1) [35–38]. All three countries have young population age structures, with more than 40% of the population aged below 15 years and are "normatively conservative" (p.13) concerning unmarried adolescent sexuality [39]. In addition to limited services, adolescent use of SRH services is

**Table 1. Legal and policy contexts for adolescent abortion, by country.**

|  | Ethiopia | Malawi | Zambia |
|---|---|---|---|
| **Legal exemptions for abortion** | Life, mental and physical health of pregnant woman; rape and incest [based on an individual's report without requiring medical or police evidence]; mental or physical disability including due to minority status of pregnant woman; foetal impairment. Includes provision to terminate pregnancies legally on the grounds of being below the age of 18 without requiring proof of age. | Life of pregnant woman | Life, mental and physical health of pregnant woman; physical and mental health of existing children; foetal impairment; adolescents under 16 |
| **Availability of safe abortion services** | Widely available in the public, private and NGO sectors. Conscientious objection is not permitted for public or private providers. | Very limited availability; no services in public sector facilities. | Some availability in public sector facilities; limited availability in the private/ NGO sector and limited reliable information about access. Individual health-care providers who have objected are required to refer the woman to another provider. |

Sources: [39,45,46]

constrained in all three countries by concerns including confidentiality, abortion stigma, provider biases, negative public attitudes about abortion and low levels of knowledge [27,37,40–44].

A review of the legal frameworks (consent laws, abortion laws and contraception access laws) in the three countries' laws and policies showed differing conceptualisation of adolescent reproductive autonomy by the State and judicial branches [39]. Zambia's abortion law is the "most progressive" [39] of the three because it includes the physical and mental health of existing children of the pregnant woman and account may be taken of the pregnant women's environment and age [35,38]. Abortions can only be provided in registered facilities and three medical doctors, one of whom must be a specialist, must be signatories for the procedure [47]. Ethiopia reformed its abortion law in 2005, maintaining abortion as illegal but incorporating substantial exceptions to abortion access without requiring legal proof or burdensome documentation from clients or health workers, including in cases where the pregnant woman is a minor. For adolescents, Ethiopian law is therefore more expansive than Zambia because it explicitly considers adolescent-specific barriers and permits legal abortion on the grounds of being below the age of 18 without requiring proof of age [48–49]. Abortion penal codes were first introduced in Malawi and Zambia during British colonial regime, and Malawi's law remains little altered from its colonial origin. Legal abortion in Malawi is currently restricted to the sole indication of endangerment to the woman's life [36,50]. Ethiopia and Zambia both have national standards and guidelines (S&G) produced by the Ministry of Health for the provision of abortion and PAC; Malawi has S&G for PAC [51]. Ethiopia has wide availability of abortion services in the public, private and NGO sectors, Zambia has some availability in the public sector, while Malawi has almost no abortion service availability [17,52–54]. Ethiopia has established protocols that do not require third-party consent from parents or partners to receive abortion care. Zambia guidelines recognise adolescents as a special category, but do not include specific protocols [39].

## Methodology

The evidence we analyse in this paper is drawn from post-procedure interviews conducted in health facilities with adolescents 10–19 years old seeking either safe abortion (SA) or post-abortion care (PAC) for treatment of complications following an abortion initiated elsewhere. Adolescents seeking care at urban public sector facilities were recruited; a total of 313 interviews were conducted (Ethiopia N = 99 [19-04-2019 – 17-09-2019]; Malawi N = 104 [14-05-2018 – 04-05-2019]; Zambia N = 110 [05-04-2018 – 25-09-2019]). Our approach to sampling was purposive and facility-based, with recruitment from urban public clinics and wards that provided abortion-related care at each study facility. All adolescents identified as having sought either SA or PAC by a study-trained senior nurse were invited to participate once they were ready for discharge. Nurses were not involved in the research consent procedures. Trained female research assistants provided informed consent to potential participants, with careful consideration given to minimizing any potential re-traumatization during the process. For respondents aged below 18 years, written informed consent was sought from any accompanying parent or guardian, in addition to written assent from the respondent. Respondents aged below 18 years without an accompanying parent or guardian were considered to have the capacity to consent if they had the capacity to seek care on their own and their consent sought. This approach deploys the concept of the "emancipated minor" [55,56]. Referring nurses and research assistants were trained to assess – based on an adolescent's capacity to abortion care-seek – the capacity of emancipated minors to consent to participate in the research. Written consent was sought from all participants aged 18 years or more. To protect participants, the study information sheet did not make any mention of abortion, and participants could provide either a signature or thumbprint for assent or consent. For all participants, research assistants were trained to signpost or connect participants to appropriate services, dependent on context (e.g.: hospital social worker, One Stop Shop for survivors of violence), in partnership with the referring nurse. Data security measures included: hard copy consent/assent forms kept in a locked cabinet in a locked office; deletion of audio recordings once uploaded to an encrypted and password-protected laptop; post-study deletion of all files from study laptops; and use of a secure service for transfer of files to an encrypted server.

We knew that attempting to gain disclosure of criminalised and/or less safe abortion methods would be difficult because adolescents are afraid of admitting to behaviours that might be potentially criminalised, with punitive outcomes (police involvement, abuse from health professionals, abuse from parents or partners who may not know of their abortion decisions) and are highly stigmatized. The interview environment was designed to ensure that adolescent safety, privacy and wellbeing were centered. Interviews were conducted in private offices. Interview questions were designed to elicit quantitative and qualitative evidence and included sexual and reproductive health (SRH) behaviours, pregnancy awareness and confirmation, care-seeking, decision-making, costs, attitudes, satisfaction with services and knowledge. Context-specific visual prompts were developed as a way of helping to accurately identify the variety of medications and local abortifacients (real and perceived). The research instruments can be found at https://abortioninafrica.wordpress.com/.

In our analyses, 'abortion safety' (safe versus less or least safe) was determined using biomedical classifications [1] and classified for each participant based on their interview self-reports and medical records (where available at time of interview). 'Safe' abortions were classified as those for which adolescents received medically recommended care (medication or procedural abortion) within a clinical setting. Self-use of medication abortion was categorized as 'less safe' and was based on adolescent's descriptions of their experiences and the drugs they used. "Least safe" abortions included the use of all other (non-medication abortion) pharmaceuticals and/or other toxins, objects inserted in the uterus, or herbal abortifacients. Descriptive statistics were analysed using Stata [57] and statistical significance defined as $p < 0.05$.

Themes were explored in the interviews using both quantitative and qualitative interviewing techniques using a two-interviewer approach [52]. Qualitative data were coded, and content analysed in Dedoose software by a team of five [EC, MC, TF, RG, JS], using a combination of deductive and inductive themes to explore issues related to barriers and facilitators of adolescent abortion and contraceptive care-seeking and decision-making in each country. Verbatim translation and transcription of the recorded interviews were conducted by the research assistants who conducted the interviews. To ensure shared understanding of research domains, coding was developed collaboratively, including blind coding by five team members of 10 cases each to check for intercoder variability and understandings. Nearly half (49%) of the interviews were coded by two members of the team, who were blinded to the other coder to check for internal consistency; the remaining interviews were single coded. Coded transcripts were re-read multiple times by team members [EC, TF, MC, JS] for analyses.

Ethical review was obtained in Ethiopia (St Paul's Hospital Millennium Medical College: PM23/146), Malawi (National Health Sciences Research Committee: 2003), Zambia (ERES: 2017-Nov-005) and the UK (London School of Economics: 000606).

## Participants

Within the sample, adolescents seeking abortion care were generally older in Ethiopia than in Malawi and Zambia, and more likely to be engaged in some form of paid work (Table 2). In Malawi, over 60% of adolescents interviewed were aged under 18. Although the majority (60%) of adolescents in Ethiopia were no longer attending school, reflecting the older age distribution of the Ethiopian sample, most adolescents in Malawi (70%) and Zambia (73%) were still in school. Two adolescents – both from Ethiopia - had never attended school. Over a fifth of adolescents in all countries were neither at school nor working. A minority of adolescents in each country were either married, cohabiting with, or were divorced, or separated (16% in Ethiopia, 15% in Malawi, and 6% in Zambia). There were no statistically significant associations between age or marital status and country. The association between country and whether an adolescent was currently attending school and/or doing paid work was significant $X^2_{(6)}$, N = 313) = 57.48, $p < 0.001$. In Ethiopia over a third (35%) of adolescents reported currently doing paid work and were not in school, compared to 9% and 5% in Malawi and Zambia, respectively.

## Results

Adolescent abortion-related care trajectories and abortion safety varied across countries. Nearly all adolescents in Ethiopia (98%) obtained a safe abortion, with most adolescents in Zambia and almost all adolescents in Malawi obtaining a less or least safe abortion (66% and 96%, respectively). Adolescents offered diverse and often multiple explanations for

**Table 2. Socio-demographic characteristics of study participants, by country.**

| | Ethiopia (n = 99) | | Malawi (n = 104) | | Zambia (n = 110) | |
|---|---|---|---|---|---|---|
| | N | % | N | % | n | % |
| **Age<sup>n.s.</sup>** | | | | | | |
| 10-14 | 0 | 0% | 4 | 4% | 12 | 11% |
| 15-17 | 28 | 28% | 61 | 59% | 53 | 48% |
| 18-19 | 71 | 72% | 39 | 38% | 45 | 41% |
| **Marital status <sup>n.s.</sup>** | | | | | | |
| Never-married | 83 | 84% | 89 | 86% | 104 | 95% |
| Currently married/cohabiting | 5 | 5% | 8 | 8% | 5 | 4% |
| Divorced/separated | 11 | 11% | 7 | 7% | 1 | 1% |
| **School and paid work[a] *** ** | | | | | | |
| Currently in school and paid work | 12 | 12% | 7 | 7% | 4 | 4% |
| Currently in school, no paid work | 27 | 27% | 65 | 63% | 76 | 69% |
| Currently has paid work, not in school | 35 | 35% | 9 | 9% | 6 | 5% |
| Not currently in paid work or school | 25 | 25% | 23 | 22% | 24 | 22% |

n.s. = not statistically significant. *** p < 0.001.

a Paid work as self-defined by respondent.

their abortion-related care-seeking, most frequently related to education and the consequences of non-marital parenthood (stigma, economic resources, familial reaction). Sexual violence was explicitly mentioned in adolescent's accounts in all three countries, ranging from 26% (n = 26) of adolescents in Ethiopia, to 19% (n = 21) in Zambia to 7% (n = 7) in Malawi. Given the well-established under-reporting of sexual violence, these figures are likely to be lower than the actual experiences of sexual violence in our sample. Perpetrators of violence included family members, boy/friends and strangers; in Ethiopia, employers were also perpetrators of sexual violence.

Before considering trajectories, delays, and experiences in detail, we note that most adolescents spoke about abortion in positive affective terms – drawing explicit links to their future plans and ambitions:

> "If I was still pregnant, I might have committed suicide. I wouldn't have stayed with my family being pregnant like this. I used to be overwhelmed. So, I was not happy. But now, that has changed." [Ethiopia, 15y]

> "I feel like a weight has been lifted off me" [Malawi, 19y]

> "[…] I want to go to school, I don't want to play with my life" [Zambia, 17y]

Across all countries, future goals and aspirations were prominent in adolescents' reflections; the consequences of not having an abortion ranged from having to leasve school, to losing sight of their future goals and the possibility to improve their livelihoods, to considering suicide. Although adolescents frequently expressed normative and stigmatising views about abortion, their own and those of others, for nearly all the participants in the study, the predominant feeling they expressed post-abortion was that of relief. For some adolescents, treatment of the complications of their less/least safe abortion contributed to their feelings of relief.

## Adolescent trajectories to abortion care

Cross-country variations in the relative safety of the abortion and type of care sought by adolescents in our study mirror laws, policies and service availability; from wide availability to adolescents in Ethiopia, to quasi-legality and limited availability in Zambia, and to a public sector abortion service that is almost entirely restricted and unavailable in Malawi (Fig 1).

PLOS Global Public Health

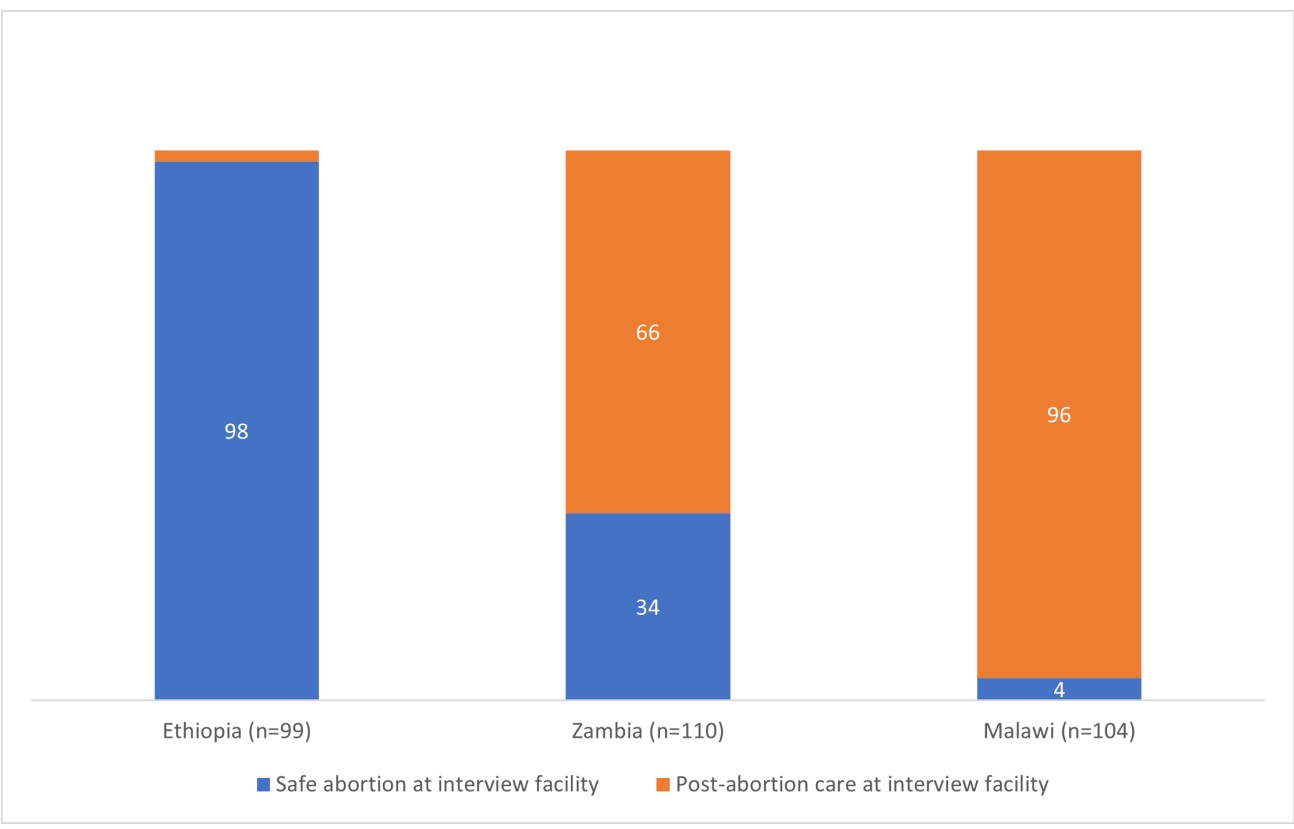

**Fig 1. Type of abortion-related care sought by adolescents at study facility, by country.**

Underpinning these country-level variations in care-seeking are the abortion trajectories of adolescents (Fig 2) that reflect the differing abortion policies and legislation in each context. The most facility-based abortions were recorded where care is most accessible (Ethiopia), and most non-facility-based and least safe abortions were recorded where care is the most restricted (Malawi). A higher proportion of adolescents in Zambia and Malawi attempted to induce an abortion more than once, compared to adolescents in Ethiopia, almost all of whom sought safe abortions in a health facility and required no known follow-up care.

Non-facility medication abortion (MA) included mifepristone and misoprostol or misoprostol only (quantity or other regimen details unknown). Non-MA pharmaceuticals that adolescents reported using included: chloroquine, pain killers, antibiotics, HSV medication, malaria tablets, vitamins, and (unspecified) vaccine/ injection. The category of "Other" included a wide range of materials and methods, either used singly or in combination, including: aloe vera, rat poison, cassava stick, guava leaves, pineapple, herbs, Coca-Cola, alcohol, pepper, stick(s), soot, inserted finger(s), tree stem, and soap powder.

In all three countries, adolescents emphasised that they were prepared to do anything necessary for their abortion. This resolve to resist the continuation of a pregnancy, despite significant obstacles – including an imperative to retain secrecy for many, underpins adolescents' abortion trajectories and abortion methods.

"[…] I made up my mind to abort, so I was ready to risk it all whether it meant dying." [Malawi, 18y]

"He [healthcare worker] told me to come to the clinic and said that such things should not be shared with people but should just be kept in secret. That is when I came here, and he removed it." [Zambia, 19y]

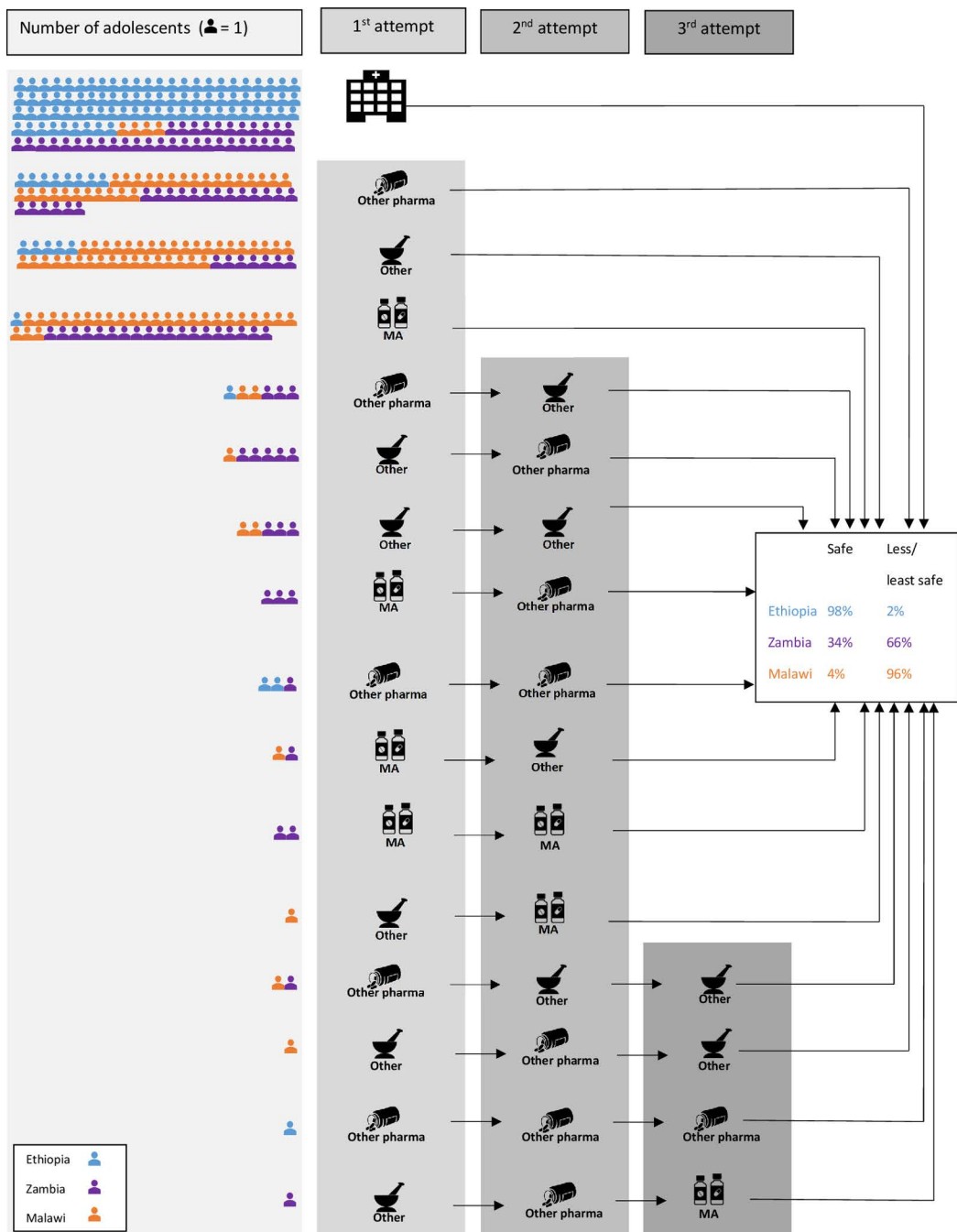

**Fig 2. Number of adolescent care-seeking trajectories and approximated safety, by country.**

Int1: Were you afraid of harm when you took it?

R:   I never thought of any effects. I would be happy if I had died at that time. [Ethiopia, 18y]

Despite differing policy environments, adolescents in all three countries sought to self-manage abortion using less-or least-safe methods. In Malawi nearly all (96%) adolescents self-managed their abortion, of which the majority (74%) were

least safe, and subsequently sought facility-based post-abortion care. In Zambia, despite five decades since the Termination of Pregnancy Act and some availability of abortion care in the public sector, two-thirds (66%) of adolescents in our study self-managed their abortion followed by post-abortion care. In Ethiopia, although 98% of the sample had a safe abortion at a facility, nearly a fifth (18%) of adolescents first tried to self-manage an abortion. Adolescents in our study self-managed – or attempted to self-manage - abortion one to three times using a range of methods that may have been dangerous, toxic, ineffective, or used incorrectly, including: self-managed medication abortion (SMMA: misoprostol or a combination of mifepristone and misoprostol); non-medication abortion pharmaceuticals (e.g.: chloroquine, antibiotics, malaria tablets, painkillers); and, other non-pharmaceutical methods (e.g.: insertion of foreign object into uterus, soap powder, alcohol, aloe vera).

In all three countries, for those adolescents that were able to disclose their need for an abortion, some were supported – information, resources, transport – by a range of people (friends, family, employer, neighbour, sexual partner). In their decision-making about the risks of different abortion methods, adolescents considered broader social safety – police involvement, familial anger, stigma – more often than medical safety:

> "I feared that I might die, or the stick inside might damage some organs […] I was afraid my family will be angry with me [for being pregnant]." [Malawi, 15y]

> Int: So before going to the drugstore, did you not know that you could abort from the hospital?

> R: I knew, but I was really scared

> Int: Why were you scared?

> R: If you ask anyone, they will tell you that if you go to the hospital seeking abortion services, they will take you to the police. [Zambia, 15y]

> "I was not that afraid [of taking pills], actually. I was sure that either it would terminate the pregnancy, or it would kill me, and I preferred dying than my family hearing what happened and get angry with me." [Ethiopia, 19y]

Self-managed medication abortion (SMMA) is a safe and effective method when correctly used [30]. However, knowledge and availability of MA pharmaceuticals and how to use them is still limited for adolescents in these countries. The combined MA regimen of mifepristone and misoprostol, with the highest efficacy rate and the simplest packaging instructions, is only registered and available in Ethiopia and Zambia and often not available without a prescription outside of a health facility [45,54]. There is a significant association between country and whether an adolescent tried to end a pregnancy before coming to the facility, $X^2(2, N = 313) = 135.93$, $p < 0.001$. In Zambia, over one-third (38%) of adolescents who tried to self-manage an abortion outside a health facility used SMMA, most commonly mifepristone and misoprostol. In Malawi, over a third (39%) of all self-managed abortions involved medication abortion, all of which were misoprostol only, which often requires multiple dosages. In contrast, in Ethiopia, only one adolescent self-managed a medication abortion, using misoprostol only (Table 3). SMMA with mifepristone and misoprostol was only reported in Zambia.

The number of different attempts that adolescents made to have an abortion also reflects the legal and health system context, and levels of knowledge about and accessibility of different methods. In Ethiopia, of the minority of adolescents that tried to self-manage an abortion before going to a facility, most made only one attempt. In Malawi and Zambia, 8% and 25%, respectively, made two or three attempts. Given the time-critical nature of abortion care, multiple attempts can reflect adolescents' increasing urgency to end the pregnancy. The initial decision-making often involved in abortion care-seeking as well as each subsequent abortion attempt resulted in increasing delays. A minority of adolescents in all three countries reported that their abortion was coerced or done to them without their consent by a third party (e.g.: boyfriend, husband, other relative); these analyses are reported in detail elsewhere [20].

**Table 3. Distribution of pre-facility abortion attempts by method, and country.**

| | Ethiopia (n=99) | | Zambia (n=110) | | Malawi (n=104) | |
|---|---|---|---|---|---|---|
| | n | % | N | % | n | % |
| Tried anything to end pregnancy before facility*** | 18 | 18% | 71 | 65% | 101 | 97% |
| For those that tried to end pregnancy before facility, what tried* | | | | | | |
| MA: mifepristone+misoprostol | 0 | 0% | 22 | 31% | 0 | 0% |
| MA: misoprostol only | 1 | 6% | 12 | 17% | 39 | 39% |
| Non-MA pharmaceuticals | 14 | 78% | 27 | 38% | 27 | 27% |
| Other | 8 | 44% | 32 | 54% | 47 | 47% |

*** p<0.001.

Note: Distribution of method of attempt sums to greater than the number that tried anything to end pregnancy before facility because some adolescents made up to three attempts (Fig 1).

### Delays in care-seeking and care-receiving

Time since pregnancy recognition and the decision to have an abortion determines how long someone has to seek and procure an abortion as procedures become more costly, less available, more complex, and limited due to national policies and practice as weeks of pregnancy proceed. Fig 1 shows the complexity of adolescent abortion-care seeking; this representation does not include time to complete a trajectory, or the delays encountered along the way. In all three countries, adolescents experienced delays to their care-seeking; in Ethiopia the proportion who reported delayed care-seeking (39%) was much lower than in Malawi and Zambia (74% and 73%, respectively) and there is a significant association between country and delay in care-seeking, $X^2(2, N=294) = 35.93$, $p<0.001$ (Table 4). Experience of delay was highest in Malawi (71%), followed by Zambia (66%), and then Ethiopia (39%). The causes and experiences of those delays vary between the three countries. There were multiple, sometimes concurrent, causes of delays to adolescent abortion care-seeking.

In Malawi, a fifth of adolescents that gave a reason for delayed care-seeking said that being "afraid" (e.g.: of family, of consequences) was the cause of their delay. Poverty also contributed to delayed care:

"My mother pleaded that they [private provider] treat us, and we pay later, but they refused." [Malawi, 18y].

Int: Did you came here right away, or you delayed after being given the referral?

R: The doctor said I should come here the same day but due to lack of transport money I have come today. [Malawi, 18y]

Int: So, when did you decide to abort the pregnancy; was it immediately after you realised that you were pregnant, or you delayed?

R: Just after I realised that I was pregnant, but I did not have money.

Int: So, that is why you delayed in terminating the pregnancy?

R: Yes. [Malawi, 18y]

Even with a progressive law in Ethiopia, barriers to timely abortion care persist and "not knowing where to go" was the single most common reason that adolescents gave; adolescents in Ethiopia – reflecting the relatively better overall service provision – were also delayed because of referrals or required tests (e.g.,: ultrasound). Given that abortion services are available at low or no cost in the Ethiopian public health system, this points to important information gaps in connecting

**Table 4. Delays to care-seeking, by country.**

| | Ethiopia (n = 99) | | Zambia (n = 110) | | Malawi (n = 104) | |
|---|---|---|---|---|---|---|
| | n | % | N | % | n | % |
| **Abortion care-seeking***** | | | | | | |
| Sought care immediately | 60 | 61% | 26 | 24% | 22 | 21% |
| Delayed | 39 | 39% | 73 | 66% | 74 | 71% |
| Missing[a] | 0 | 0% | 11 | 10% | 8 | 8% |
| **If delayed, duration (days)** | | | | | | |
| Min | 2.0 | | 1.0 | | 1.0 | |
| Max | 90.0 | | 90.0 | | 120.0 | |
| Median (Interquartile range) | 14.0 (7.0-30.0) | | 7.0 (3.0-14.0) | | 14.0 (3.5-60.0) | |
| **Reason(s) given for delay (number of responses)** | | | | | | |
| | 1st, 2nd or 3rd reason given | | 1st, 2nd or 3rd reason given | | 1st, 2nd or 3rd reason given | |
| Other[b] | 31 | | 54 | | 42 | |
| Was afraid | 4 | | 12 | | 15 | |
| Did not know what to do | 1 | | 6 | | 11 | |
| Did not know where to go | 10 | | 4 | | 3 | |
| Did not have transportation | 1 | | 3 | | 2 | |
| Partner refused | 0 | | 2 | | 1 | |
| Tried to go to other facility, turned away | 2 | | 0 | | 0 | |
| Missing | 0 | | 11 | | 5 | |

*** p < 0.001.

[a]Missing data included respondents that: did not provide this information; that reported an abortion being done without their knowledge (e.g.: being given medication abortion drugs on the pretext that they were medicine to support a pregnancy).

[b]Other reasons for delay: Had to attend a funeral; delays in pregnancy recognition; time to persuade or negotiate with others (e.g.: boyfriend, mother); time to gather resources (e.g.: financial); could not miss school/ college/ employment (including waiting for school/ college term to end); time to identify an adult to accompany her; trying to find abortion method (e.g.: medication abortion) or provider (e.g.: doctor, pharmacist), including multiple attempts to identify a source; time to care-seek in secret (including to be able to travel whilst bleeding); healthcare workers required third party consent (e.g.: boyfriend); waiting for police response (for a rape case with police involvement); indecision (own and/ or third party) about whether to have an abortion; denial of care.

Note: Responses do not sum to the full sample sizes since multiple responses were possible.

those services to the people who want to use them. In some cases, healthcare workers turned away adolescents due to negative attitudes towards adolescent sexual activity:

"It has been six days since I first find out about the pregnancy. I asked the health professionals in that health centre to terminate the pregnancy. But the nurse there said that I should have protected myself [rather] than going there to get an abortion service. I just kept quiet and got out of there." [Ethiopia, 18y]

Potential unknown risks – reflecting the vacuum of information available to most adolescents – contributed to delays as adolescents weighed up the possible risks of abortion against their desire to terminate a pregnancy:

Int: ...why did you delay doing something?

R: Mum, I was thinking about it because I was scared, because people say, when aborting, it is a matter of dying and surviving so I was still thinking about it. [Zambia, 18y]

Although abortion-related care is available at low or no cost in the public sector in Zambia, adolescents incurred delays often related to gathering funds for private providers, herbalists, or transport:

> "That is when I delayed for like four days, then I think it [the pregnancy] became 11 weeks. I came here and they told me to go for a scan. [Pause] Now I never had money, the money for a scan, so I just took the medicine the man gave me." [Zambia, 17y]

Adolescents described and explained delays that were less about the decision to have an abortion, and more about the structural factors they faced (e.g.: inadequate information, time to procure financial resources). Adolescents in Ethiopia were much less likely to report being afraid as a reason for delaying care-seeking; this likely reflects the higher availability of abortion care in Ethiopia and that abortion is less stigmatised than in Malawi and Zambia.

### Adolescents' experiences in health facilities

Quality healthcare experiences are multifaceted and include the need for effective communication, respect and dignity, and emotional support [30]. In response to direct questions, adolescents reported high levels of respectful treatment across all three countries (Table 5). The majority felt that they were made to feel welcome, and that the healthcare provider was polite. There were significant associations between country and whether an adolescent was provided with an emergency phone number ($X^2$(2, N = 309) = 18.91, $p < 0.001$) with 27% of adolescents in Zambia receiving a phone number compared to 6% of adolescents in Malawi or information about subsequent return to fertility ($X^2$(2, N = 306) = 27.94, $p < 0.001$) with 71% of adolescents in Ethiopia informed about return to fertility compared to 36% of adolescents in Malawi.. There was no statistically significant association between country and whether the treatment was respectful, the adolescent was made to feel welcome, the health provider was polite or an unofficial payment was made to staff. Qualitative evidence similarly indicated that many adolescents felt that the care was respectful and that they were treated well:

> "[…] I came here for bad thing. I mean it is not acceptable. But they didn't accept me on that way.
>
> They treated me normally and made me not to fear." [Ethiopia, 19y]
>
> "They welcome me, nicely. They smile at me as if am their child." [Zambia, 17y]

In the first extract above the adolescent offers insights into her understandings of social norms about abortion as a "bad thing" that is "not acceptable" and is illustrative of how expectations of quality abortion care might be low.

However, responses to other probing questions belie more complex, nuanced, and often negative, experiences that a smaller number of adolescents spoke about contributing to their feelings of shame. Adolescents reported how they were stigmatised for being pregnant, which was reflected in their experiences of care:

> "They [healthcare workers] just scolded me, they wanted a plastic [sheet] and they asked if we bought a plastic and we said no […] They were scolding me saying I shouldn't be pregnant, am young and so on and I should continue with school." [Zambia, 16y]
>
> Int: Why do you think they [medical officer] were not talking to you politely? What did they do?
>
> R: They did that because I did a bad thing. [Malawi, 17y]

An adolescent from Zambia who experienced disrespectful care offered her thoughts on how healthcare workers' training could be improved:

**Table 5. Experiences of facility-based abortion-related care, by country.**

| | Ethiopia (n = 99) | | Zambia (n = 110) | | Malawi (n = 104) | |
|---|---|---|---|---|---|---|
| | N | % | n | % | N | % |
| **Treatment was respectful** n.s. | | | | | | |
| Yes | 94 | 95% | 107 | 97% | 100 | 96% |
| No | 5 | 5% | 2 | 2% | 1 | 1% |
| Did not answer | 0 | 0% | 1 | 1% | 3 | 3% |
| **Made to feel welcome** n.s. | | | | | | |
| Yes | 93 | 94% | 106 | 96% | 98 | 94% |
| No | 6 | 6% | 3 | 3% | 3 | 3% |
| Did not answer | 0 | 0% | 1 | 1% | 3 | 3% |
| **Health provider was polite** n.s. | | | | | | |
| Yes | 95 | 96% | 106 | 96% | 96 | 92% |
| No | 4 | 4% | 3 | 3% | 3 | 3% |
| Did not answer | 0 | 0% | 1 | 1% | 5 | 5% |
| **Given emergency phone number*** | | | | | | |
| Yes | 6 | 6% | 30 | 27% | 13 | 13% |
| No | 93 | 94% | 79 | 72% | 88 | 85% |
| Did not answer | 0 | 0% | 1 | 1% | 3 | 2% |
| **Given information about return to fertility*** | | | | | | |
| Yes | 36 | 36% | 57 | 52% | 74 | 71% |
| No | 62 | 63% | 51 | 46% | 26 | 25% |
| Did not answer | 1 | 1% | 2 | 2% | 4 | 4% |
| **Unofficial payment made to staff** n.s. | | | | | | |
| Yes | 7 | 7% | 12 | 11% | 3 | 3% |
| No | 92 | 93% | 94 | 85% | 91 | 88% |
| Don't know | 0 | 0% | 2 | 2% | 0 | 0% |
| Missing | 0 | 0% | 2 | 2% | 10 | 10% |
| Range of amounts paid (US$) [mean ($\bar{x}$), standard error(SE)] | 0.9-24.6 [$\bar{x}$ =12.73, SE=6.94] | | 4.2-125.9 [$\bar{x}$=39.22, SE=14.75] | | 1.0-2.7 [$\bar{x}$=1.59, SE=0.51] | |

n.s.= not statistically significant. *** $p < 0.001$.

"But when they were training, being trained to be nurses, were they told they would handle acertain age group? […] So, I think that is really negative...a girl is a girl. Whether 14, whether 10...Sothat is what really hurt me listening to the comments....I told my aunt I don't want to receive themedication, better I go die from home and then you come and bring me here as a dead body"[Zambia, 18y]

Adolescents' affective responses to the care they received included feelings of guilt:

"[…] when they said it was my fault, I really felt bad since I was scared before about what people said to me. If they treated us better, we didn't [wouldn't] lose our time by thinking guilty about ourself." [Ethiopia, 19y]

Healthcare workers drew on normative scripts about education and marriage in their reported attitudes towards unmarried schoolgirls seeking abortion-related care:

Int: Ok, so can you explain to me what happened when they were shouting at you?

R: They were saying that I am supposed to finish school first, then go college and marry a guy. [Zambia, 18y]

Adolescents also experienced refusal of care – including for care following sexual violence – from some healthcare workers:

"[…] the doctor refused saying that he cannot give me the medicine, saying that he cannot give me because what I want to do is wrong, that I should just keep my pregnancy […] he said he cannot give me the medicine because it is more like he is encouraging me to go ahead with what I want to do. He left the room, the person who gave me is someone else." [Zambia, 17y]

A minority of adolescents in each country reported that they – or someone accompanying them – had paid unofficial money to facility staff (Table 5). Such payments were most frequently reported in Zambia (10% of respondents) compared to Malawi and Ethiopia (3% and 7%, respectively).

### Post-abortion contraception

Counselling about contraception is a standard component of post-abortion care (WHO 2022), and adolescents' responses to questions about post-abortion contraception give important insights into their experiences. There were significant associations between country and whether an adolescent was offered ($X^2(2, N = 312) = 66.95$, $p < 0.001$) or wanted ($X^2(2, N = 266) = 26.21$, $p < 0.001$) post-abortion contraception. The association between country and whether an adolescent reported receiving contraception was not statistically significant. There were different experiences of post-abortion contraception in the three countries. In Ethiopia, 84% of adolescents reported being offered contraception, and 80% said that they

**Table 6. Post-abortion contraception, by country.**

| | Ethiopia (n = 99) | | Zambia (n = 110) | | Malawi (n = 104) | |
|---|---|---|---|---|---|---|
| | n | % | n | % | N | % |
| Offered contraception?*** | | | | | | |
| Yes | 83 | 84% | 33 | 30% | 41 | 39% |
| No | 16 | 16% | 76 | 69% | 63 | 61% |
| Missing | 0 | 0% | 1 | 1% | 0 | 0% |
| Wanted contraception?*** | | | | | | |
| Yes | 64 | 65% | 32 | 38% | 44 | 42% |
| No | 18 | 18% | 59 | 54% | 39 | 38% |
| Missing[a] | 17 | 17% | 9 | 8% | 21 | 20% |
| Received contraception? n.s. | | | | | | |
| Yes | 79 | 80% | 14 | 13% | 30 | 29% |
| No | 20 | 20% | 93 | 85% | 69 | 66% |
| Missing | 0 | 0% | 3 | 3% | 5 | 5% |
| % of those not offered contraception who reported wanting contraception | 31% (n = 5) | | 38% (n = 29) | | 51% (n = 32) | |
| % of those that received contraception who reported initially not wanting contraception | 5% (n = 3) | | 0% | | 7% (n = 2) | |

n.s. = not statistically significant. *** p < 0.001.

a Missing category includes no response and responses that were ambiguous (e.g.: adolescent described "accepting" contraception, rather than "wanting").

had "accepted" contraception, although only 64% reported initially wanting family planning, a potential sign of coercion (Table 6). In comparison, in Zambia, 30% of adolescents reported being offered family planning, while 38% reported wanting family planning, and only 13% ultimately received a family planning method as part of their post-abortion care. In Malawi, 39% of adolescents reported being offered family planning, with 42% wanting family planning and only 29% receiving it.

There were similarities in terms of the family planning methods offered. In all three countries, substantial proportions of those not offered contraception reported wanting contraception; in Malawi, this was more than half (51%) of adolescents who reported not being offered post-abortion contraception. Of those that received post-abortion contraception in Ethiopia, nearly all (99%) reported a LARC (implant – 91%, injectable – 8%); rates of reported LARC post-abortion contraception were similarly high in Malawi and Zambia (76% - 3% implant and 73% injectable, 85% - 14% implant and 71% injectable, respectively). A minority of adolescents in Ethiopia and Malawi reported having received contraception, despite not wanting it (5% and 7%, respectively).

Our qualitative evidence shows a range of experiences including an absence of choice of contraceptive method:

Int: They explained to you about the family planning methods?

R: Yes.

Int: And you have accepted to be given the family planning methods?

R: Yes, they told me about contraceptive pills, but they said they were out of stock.

Int: So, which other method will you prefer?

R: The pills

Int: You said that you were given injection was the injection not for family planning?

R: Mmmh I don't know.

Int: You should have asked, the use of the injection.

R: I was also surprised but I never asked.

Int: Alright but you would have loved if you used the pills?

R: Yes. [Malawi, 19y]

Int: So, have you been given contraception methods today?

R: They just told me that I need to be getting an injection. [Malawi, 18y]

An absence of choice and experiences of coercion were reported most frequently in Ethiopia; for some adolescents, contraception was perceived as a requirement to be able to access abortion services:

"I don't want it. A nurse told me that I have to use contraception for the first time when I registered at {study site} and told me for the second time when I was in the ward. But I said to her that I don't want it. She couldn't understand me, she considered me as a rude girl and treated me badly [crying]." [Ethiopia, 18y]

Int  Were you offered a family planning method today?

R: Yes.

Int: Which type of method?

R: They told me as it works for three years.

Int: Did they tell you about another option?

R: No, they did not tell me.

Int: Was it your choice?

R: No, it was not my choice. I thought that they would not provide me the service, or the pregnancy would not terminate if I did not use family planning then I accepted it.

[Ethiopia, 18y]

## Discussion

Adolescents' capacities to achieve reproductive autonomy are constrained or enabled across three contrasting legal contexts; many of our results correspond with the level of restrictions on legal abortion in the three countries. Our comparative analyses demonstrate how individual adolescent abortion care-seeking are impacted by meso- and macro- or structural-level relationships and systems [58] 2019). These "often unnoticeable" systems – including legal systems - shape an individual's choices and behaviours and are a form of structural violence [59–60]. A highly restrictive legal framework and very limited legal abortion services in Malawi underpins lengthy abortion trajectories, ending in less safe abortion for nearly all the adolescents in our sample. Because mifepristone is not registered or available in Malawi (as it is in Ethiopia and Zambia), options for MA are limited to misoprostol but the restrictive legal environment limits knowledge and discussion of, and access to, misoprostol that could result in safe and effective self-managed abortion [8,36]. A more permissive legal and service delivery framework in Ethiopia – with explicit provision in the Penal Code for legal minors – shapes abortion trajectories that are safer for nearly all (98%) adolescents. For the minority of Ethiopian adolescents who delayed care-seeking or were trying to manage their own abortions, their trajectories to care were relatively longer than those who sought facility-based care. In Zambia, despite longstanding legal provision for abortion, the majority of adolescent trajectories involve seeking post-abortion care for an abortion initiated elsewhere, highlighting that legal provision on its own does not necessarily translate into access. In all three countries, adolescents' articulations of the potential risks that their abortion-seeking involved serve to emphasise the strength of their resistance to constraints on their individual reproductive autonomy.

Many of the adolescents who spoke with us had exercised agency – to the extent possible – prioritising their own needs; yet in doing so they sometimes put their lives at risk, using least safe methods. By focusing on the non-linear sequences of events in a single trajectory from the seeking of care to the experience of abortion care and then post-abortion contraception, we show how, cumulatively, institutions cause harm to adolescents and further restrict their limited autonomy. Eliciting detailed information about each part of a trajectory – Was anybody else involved? Why? How much did it cost? Who paid? etc. – offers insights into the ways in which adolescents' constrained access to financial resources played a role in their care-seeking. For many adolescents the time needed to gather financial resources (for transport, fees, etc.) contributed to delays in care-seeking. Understanding adolescent abortion care-seeking trajectories exposes the nuanced ways that age and gender intersect and collide with structural systems and institutions in all three countries. For many adolescents, the decision to have an abortion – and then acting on that decision – is an act of resistance. Our work shows that it is possible – with limitations (see below) - to generate evidence about adolescent abortion-seeking, including from very young adolescents. Future research could consider the severity of abortion complications, and the mental health implications of abortion care-seeking for adolescents.

Individual experiences of abortion care and care-seeking – each critically important in their own right – cannot be made sense of without being situated in their broader structural context. Negative attitudes towards adolescent sexuality are backed by the law, including penal consequences for consensual sex, which exacerbate stigma [61]. Concurrent and often mutually reinforcing structural factors (e.g.: legal, health systems, social and cultural norms) act to shape the difficult

conditions within which adolescents make and then act upon a decision to have an abortion, with implications for the type and safety of abortion. For example, societal stigmatisation of (unmarried) adolescent sexuality, pregnancy, and abortion – to varying extents - in all three countries [40,62,63] meant that abortion trajectories were primarily shaped by secrecy, lack of support and most often a lack of reliable information, especially in Malawi and Zambia. Adolescents sought to maintain secrecy in ways that balanced their biomedical and social safety [64], fearing that people would find out about their sexual activity, with implications for complex and sometimes lengthy abortion trajectories. The need to maintain secrecy to stay in school or continue to live with parents or employers makes it more difficult to find out information, or ask questions, or approach a health facility [8,65,66]. Secrecy makes it easier for adolescents to be exploited, for example, by health-workers demanding unofficial fees or turning them away; it also put them at risk of additional morbidity due to delayed care-seeking for complications as they hid their symptoms.

Social norms – distinct from individually held beliefs or attitudes - are instrumental to understanding and meeting adolescent sexual and reproductive health needs [67–68]. Adolescents in our study invoked injunctive norms about themselves and their abortion care-seeking – that their sexuality, pregnancy, and abortion were "wrong" or "bad". Adolescents spoke about how others they encountered during their abortion care-seeking trajectories, including their health care providers, also invoked such injunctive norms concerning their age or having an abortion; by transgressing these norms, adolescents were exposed to stigmatising experiences, including disrespectful care. Our analyses suggest that education and training for healthcare workers involved in the care of adolescents could be improved with inputs focused on adolescent-centred and non-stigmatising, reflecting findings elsewhere [12,14,26,69].

Time is of critical importance in abortion care-seeking, and delay(s) have implications for whether and what type of care is possible, as well as the social and economic resources required. A review of abortion knowledge, attitudes and experiences among adolescents concluded that adolescents are more likely than older women to delay abortion-related care [13]. A study from Malawi of adolescents aged 14–19 years concluded that multiple factors including fear of parents finding out about the pregnancy and provider abuse contributed to delayed care-seeking [70]. For an adolescent who has decided to have an abortion, delays in care-seeking represent thwarted attempt(s) to seek out information, support, resources, or care and are accompanied by sometimes profound psychological distress. In our sample, median delays of 1 week (Zambia), 2 weeks (Ethiopia) or over three weeks (Malawi) show how difficult it can be for adolescents to navigate their abortion trajectories. Experiences of delays in seeking care make visible the intersections between structural and individual-level barriers. Delays were implicated in a significant proportion of abortion trajectories in Malawi and Zambia, where abortions were less available and multiple attempts to terminate the pregnancy were more common. Adolescents in Malawi, who tended to be younger, reported delaying their care seeking because of being afraid, suggesting that restrictive policy contexts create environments and conditions that exacerbate barriers to access. In Ethiopia, adolescents experienced health system delays (e.g.: referrals to other locations, appointments). Across all three countries, not knowing where to go or who to ask suggests the significant role of information in care seeking, regardless of legality.

Quality of care in abortion – its definition and measurement – is evolving. Evidence suggests that abortion patients "overwhelmingly" report high satisfaction with services – which is not the same as high quality of care – for a range of reasons including: no longer being pregnant; being able to access a service; or, low expectations of abortion care service quality [71]. Evidence from Ethiopia reports high satisfaction with comprehensive abortion care and post-abortion care among women [72–73]. Little attention has been paid to adolescents' satisfaction with abortion care or the quality of that abortion care. Evidence from providers of post-abortion care in Kenya and Ghana suggests that unmarried adolescents may receive poor quality of care [74–75]. A multi-country study of 11 African countries that assessed adolescents' satisfaction with care for abortion-related complications compared to older women showed that adolescents were 50% more likely to be consistently very satisfied with their overall care than older women; the authors suggest that "Leaving the facility health, alive, and without pregnancy may have increased their satisfaction" [69]. Adolescents' responses in our study to closed questions about the quality of care they experienced suggest similarly overall positive experiences in

public health facilities. Adolescents with positive experiences of care show what is possible in highly constrained resource and restrictive legal settings. Some of the adolescents that we interviewed had positive abortion care-seeking experiences with healthcare workers that took account of adolescents' autonomy and decision-making. For some adolescents, their relatively young age meant that healthcare workers treated them with empathy and care. However, the dissonance between survey-type positive responses and adolescents' qualitative insights suggests that many adolescents' abortion care-seeking was difficult, stressful, and (potentially) exposing and dangerous. They feared being reported to the police, being discovered by neighbours or relatives, being made to bring their sexual partner, and disclosure of their abortion to their parents. Some healthcare workers shamed adolescents for being pregnant, regardless of whether an adolescent was pregnant because of a consensual sexual relationship or because of violence. Particularly in Malawi and Zambia, adolescents reported being interrogated about their self-management of an abortion or having threats of criminalisation used against them. Pregnant adolescents who are survivors of sexual violence face the "choice" of forced childbearing or (unsafe) abortion; becoming pregnant was not a choice. Adolescents that sought abortion-related care after sexual violence rarely mentioned any support being offered by the health workers they encountered.

Ethical and person-centred post-abortion contraceptive care includes non-coercive accessibility of a range of methods. Evidence relating to adolescents' preferences and experiences of post-abortion contraceptive care is rare [76]. Our data suggest two distinct patterns with respect to adolescent post-abortion contraception. In Ethiopia most adolescents in our study were offered (84%) and received (80%) contraception, levels that were higher than the proportion of adolescents who unambiguously stated that they initially wanted contraception (66%). Of those adolescents that received post-abortion contraception in Ethiopia, nearly all (99%) reported receiving a LARC. Our qualitative evidence suggests some Ethiopian adolescents experienced coercion and/or an absence of choice. By contrast, in Malawi and Zambia the evidence suggests that substantial proportions of adolescents would like to use contraception, but were not offered it (51% and 31%, respectively); opportunities to educate and counsel adolescents in health facilities and in countries where comprehensive sexuality education is limited were missed. In addition, some adolescents that both wanted and received contraception were offered limited contraceptive choice, reducing their contraceptive autonomy and potentially leading to future dissatisfaction and discontinuation.

Adolescents receiving abortion-related care are made vulnerable – due to their age and stigmatisation of their pregnancy and subsequent abortion – to post-abortion contraceptive care that can be unethical and not person-centred (e.g.: absence of choice, coercion to use). We speculate that the high levels of post-abortion contraceptive "acceptance" of LARC in Ethiopia might be reflective of target-driven structures and policies and/ or provider biases [27,77]. Our evidence offers insights into the concept of contraceptive autonomy for adolescent post-abortion care – the factors necessary for an adolescent to decide for themselves post-abortion what they want in relation to contraception and then to realise that decision [78]. Contraceptive autonomy sub-domains of informed, full, and free choice are absent to varying extents in our evidence across the three countries.

Our approach has limitations. Our interviews with adolescents are all based in urban areas and focus on adolescents that sought care from a public facility, many after seeking care from a wide variety of providers, locations, and facilities. Experiences of rural and/or self-managed care that did not involve or resolve in a facility and/ or adolescents that were unable to access facility-based care are excluded. Our purposive sample selection may lead to selection bias. Given the urban study sites, rural participants were underrepresented in our study, which may limit the statistical generalisability of our findings. Some rural adolescents travelled substantial distances to access care in our urban study facilities. However, adolescents in our sample were socio-economically diverse, and abortion care is geographically centralised, especially in Malawi and Zambia. Our sample is comprised of adolescents with the capabilities, agency and/ or resources to seek an abortion or post-abortion care; our approach excludes adolescents that might have wanted an abortion but were unable to seek or obtain care. Many adolescents in our study sought care following prior experiences with a wide range of providers and facilities; our sample includes diverse trajectories to reaching a public health facility. Our analyses include the

outcome of postabortion care, but do not consider the severity of complications requiring post-abortion care. Our insights about quality of care are based on adolescents' perspectives; we did not collect data from other perspectives, for example by interviewing healthcare workers about their reported practices and behaviours with respect to the provision of abortion care services. Adolescents' descriptions of the care they received offer person-centred insights into how care is experienced, and its quality. Our research focused on experiences of abortion-related care; survivors of sexual violence that had received specialist care (e.g.: One Stop centres) were not asked about their experiences of this non-abortion care. This study cannot generalise adolescent experiences in these three countries. However, we are confident that our findings are salient to the experiences of adolescents in other, similar, contexts. Finally, our data are rich but, as with all self-reported data, they are subjective and impacted by temporal issues and biases such as recall or social desirability bias.

## Conclusions

Our cross-country analysis shows that less restrictive abortion policies support adolescents to access medically safer abortions. Conversely, more restrictive laws and policies result in more less and least safe abortions. In Ethiopia, where abortion legal reform has been in effect for over 15 years, only one in five adolescents attempted to self-manage an abortion. The highly restrictive environment for abortion care in Malawi is reflected in the nearly universal attempts to self-manage abortions outside of a health facility using ineffective or unsafe methods. The complexity of the legal context, service availability and access in Zambia resulting in confusing and contradictory information is evident in the distribution and types of care-seeking among adolescents; two-thirds of adolescents interviewed sought to self-manage their own abortions or have less safe abortions outside of a health facility, despite a law that specifically refers to adolescents' right to seek safe abortion services in government facilities. When laws and health systems are miss-aligned with adolescents' abortion care needs, this exposes adolescents to multiple violences – of unsafe abortion, of exploitation, of coercive post-abortion contraceptive practices, and of disrespectful care. Adolescents' trajectories – particularly those involving multiple attempts and barriers to achieving abortion care – require tenacity and determination; this tells us important things about how important abortion care is to adolescents despite the numerous barriers they face.

## Acknowledgments

The evidence we present here could not have been possible without the time and insights of the adolescents who spoke with us. The research was conducted with the support of Isikanda Mulasikwanda and Blain Rezene. The interviews were conducted by: S. Abebe, T. Chasima, E. Chifumpu, R. Chipoya, G. Finyiza, S. Getenet, C. Kanyinji, N. Mategula, Y. Meleyo, H. Mersha, C. Mubanga, N. Mutambo, E. Tembenu, S. Weldegebriel, and F. Workneh.

## Author contributions

**Conceptualization:** Ernestina Coast, Tamara Fetters, Malvern Tatenda Chiweshe, Bellington Vwalika, Godfrey Kangaude.

**Data curation:** Ernestina Coast, Malvern Tatenda Chiweshe, Risa Griffin, Joe Strong.

**Formal analysis:** Ernestina Coast, Tamara Fetters, Malvern Tatenda Chiweshe, Risa Griffin, Joe Strong.

**Funding acquisition:** Ernestina Coast, Tamara Fetters.

**Investigation:** Ernestina Coast, Tamara Fetters, Malvern Tatenda Chiweshe, Bellington Vwalika, Luke Tembo.

**Methodology:** Ernestina Coast, Tamara Fetters, Malvern Tatenda Chiweshe, Risa Griffin.

**Project administration:** Ernestina Coast, Tamara Fetters, Malvern Tatenda Chiweshe, Bellington Vwalika, Luke Tembo, Charlotte Chishiba, Malede Birara, Grace Chiudzu, Abrham Getachew, Samuel Muluye Welelaw.

**Resources:** Ernestina Coast, Tamara Fetters, Bellington Vwalika, Luke Tembo, Charlotte Chishiba, Malede Birara, Grace Chiudzu, Abrham Getachew, Samuel Muluye Welelaw.

**Supervision:** Ernestina Coast, Tamara Fetters, Malvern Tatenda Chiweshe, Bellington Vwalika, Luke Tembo, Charlotte Chishiba, Malede Birara, Grace Chiudzu, Abrham Getachew, Samuel Muluye Welelaw.

**Validation:** Ernestina Coast, Tamara Fetters, Malvern Tatenda Chiweshe, Risa Griffin, Joe Strong.

**Visualization:** Ernestina Coast, Tamara Fetters, Malvern Tatenda Chiweshe, Risa Griffin, Joe Strong.

**Writing – original draft:** Ernestina Coast, Tamara Fetters, Malvern Tatenda Chiweshe.

**Writing – review & editing:** Ernestina Coast, Tamara Fetters, Malvern Tatenda Chiweshe, Bellington Vwalika, Risa Griffin, Luke Tembo, Joe Strong, Charlotte Chishiba, Malede Birara, Grace Chiudzu, Abrham Getachew, Samuel Muluye Welelaw, Godfrey Kangaude, Nyovani Madise.

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
