## [Decision Letter · Decision Letter 0]

6 Jan 2025

PGPH-D-24-01948

Adolescent abortion care trajectories and safety in Ethiopia, Malawi, and Zambia: a comparative mixed methods study

Dear Dr. Coast,

Thank you for submitting your manuscript to PLOS Global Public Health. After careful consideration, we feel that it has merit but does not fully meet PLOS Global Public Health’s publication criteria as it currently stands. Therefore, we invite you to submit a revised version of the manuscript that addresses the points raised during the review process.

The manuscript has been evaluated by three reviewers, and their comments are available below.

The reviewers have raised a number of major concerns. They request improvements to the reporting of methodological and statistical aspects of the study. In addition, further discussion on the limitations and ethical considerations is required.

Could you please carefully revise the manuscript to address all comments raised?

We look forward to receiving your revised manuscript.

Kind regards,

Helen Howard

Staff Editor

Journal Requirements:

Additional Editor Comments (if provided):

Reviewers' comments:

Reviewer's Responses to Questions

**Comments to the Author**

1. Does this manuscript meet PLOS Global Public Health’s publication criteria ? Is the manuscript technically sound, and do the data support the conclusions? The manuscript must describe methodologically and ethically rigorous research with conclusions that are appropriately drawn based on the data presented.

Reviewer #1: Yes

Reviewer #2: Yes

Reviewer #3: Partly

2. Has the statistical analysis been performed appropriately and rigorously?

Reviewer #1: Yes

Reviewer #2: Yes

Reviewer #3: No

3. Have the authors made all data underlying the findings in their manuscript fully available (please refer to the Data Availability Statement at the start of the manuscript PDF file)?

Reviewer #1: Yes

Reviewer #2: Yes

Reviewer #3: No

4. Is the manuscript presented in an intelligible fashion and written in standard English?

Reviewer #1: Yes

Reviewer #2: Yes

Reviewer #3: No

5. Review Comments to the Author

Reviewer #1: 1. Abstract has minor grammar and spelling mistakes

2. Verbatim on pills and post abortion contraception is not child friendly. It involves questioning a child on why she has not asked about the injection that was given to her. Please review

3. Well written manuscript.

4. Is health professions education in the study areas "gendered"? Please comment in discussion

Reviewer #2: I commend the authors for this study, which addresses critical issues in adolescent reproductive health, specifically in the context of legally diverse African countries. Its focus on abortion trajectories across Ethiopia, Malawi, and Zambia offers important insights into the effects of legal, social, and healthcare contexts on the safety of adolescent abortion experiences.

The use of a comparative mixed-methods approach adds depth to the findings, which highlight the implications of restrictive versus permissive abortion laws and the resulting health and social outcomes for adolescents, which has substantial policy relevance.

Areas for Improvement:

1. Abstract and Key Findings: The abstract could benefit from concise language and clearer presentation of key findings, particularly in terms of statistical results. Additionally, a summary of policy implications would strengthen the abstract.

2. Detailed Methodology Explanation: While the methods are robust, more details on sampling and potential biases, especially regarding urban-only recruitment, would improve transparency.

3. Data Analysis: The paper could be strengthened by including effect sizes and confidence intervals for quantitative results where relevant. This addition would help readers better gauge the practical significance of findings.

4. Discussion of Limitations: Although limitations are mentioned, expanding on how urban-centric sampling and potential recall bias may affect the generalizability of findings would enhance the study’s transparency.

5. Ethical Considerations: Greater detail on how sensitive information was safeguarded, given the vulnerability of the adolescent population, would strengthen the ethical considerations of the study.

Conclusion: This study makes a valuable contribution to understanding adolescent abortion care in varied legal contexts. With minor revisions to clarity, methodology, and ethics, this paper is suitable for publication in PLOS Global Public Health.

Reviewer #3: Dr. Coast and colleagues conducted an observational study investigating the abortion-seeking behaviors of adolescents in three countries: Ethiopia, Zambia, and Malawi. They found that, except in Ethiopia where appropriate abortion care was provided, the other two countries largely failed to provide safe abortion services. Additionally, delays in seeking abortion care were identified as a significant challenge. While these findings are not new, evidence from developing countries, particularly regarding vulnerable populations, remains valuable.

However, I suggest that the authors expand their analysis to include an investigation of the complications associated with abortion and an evaluation of the mental stress experienced by adolescents. These areas of focus would be more appropriate and would significantly enhance the quality of the article. My concerns are outlined below:

Major concerns:

1. Methods:

Would you clarify the sample formula and how did you come up with 313 cases?

Would you specify how did you chose which healthcare facilities to recruit the adolescent? and your specific effort to reduce the possible selection bias? What is the ratio of public vs. private healthcare facilities allowed to provide abortion services in those countries?

What are your inclusion and exclusion criteria?

Would you specify the definition of safety of abortion? It seems confusing for me and relies on subjects' experiences which might be very biased.

Statistical analysis section were missing.

2. Results:

Why did you quote the subjects responses? It is helpful not informative.

The results section needs to be rewrote with more appropriate, systematic, and statistical way.

3. Discussion:

This section is overwhelming due to unnecessary discussion that does not focus on the results presented in the article. I suggest that the author restructure it into 3-4 paragraphs, each addressing a specific topic: the safety and complications of abortion, delays in seeking abortion, and potential future policy recommendations.

Minor concerns:

1. Abstract:

Line 36. Typo. "....and Zambia between 2018-19l"

Lines 34-37. The methods was not well described. Would you elaborate more details about your study design including time (month/year), location (which locations have you collected the data?), inclusion and exclusion criteria...

Lines 41-49. The result might need to include the number (percentage).

2. Manuscript:

The authors might need to restructure the manuscript as Introduction - Methods - Results - Discussion - Conclusion and use subheading as needed.

6. PLOS authors have the option to publish the peer review history of their article (what does this mean? ). If published, this will include your full peer review and any attached files.

**Do you want your identity to be public for this peer review?** For information about this choice, including consent withdrawal, please see our Privacy Policy .

Reviewer #1: **Yes: ** Nancy Angeline Gnanaselvam

Reviewer #2: **Yes: ** AMOS M'YISA MAKELELE

Reviewer #3: No

---

## [Editor Report · Decision Letter 1]

14 Mar 2025

Adolescent abortion care trajectories and safety in Ethiopia, Malawi, and Zambia: a comparative mixed methods study

PGPH-D-24-01948R1

Dear Professor Coast,

We are pleased to inform you that your manuscript 'Adolescent abortion care trajectories and safety in Ethiopia, Malawi, and Zambia: a comparative mixed methods study' has been provisionally accepted for publication in PLOS Global Public Health.

Best regards,

Farzana Kapadia

Academic Editor